# Chronotype and Social Jetlag: A (Self-) Critical Review

**DOI:** 10.3390/biology8030054

**Published:** 2019-07-12

**Authors:** Till Roenneberg, Luísa K. Pilz, Giulia Zerbini, Eva C. Winnebeck

**Affiliations:** 1Institute of Medical Psychology, LMU Munich, 80336 Munich, Germany; 2Programa de Pós-Graduação em Psiquiatria e Ciências do Comportamento, Universidade Federal do Rio Grande do Sul (UFRGS), Porto Alegre 90035-003, Brazil; 3Laboratório de Cronobiologia e Sono, Hospital de Clínicas de Porto Alegre (HCPA), Universidade Federal do Rio Grande do Sul (UFRGS), Porto Alegre 90035-903, Brazil

**Keywords:** sleep-wake timing, circadian clock, entrainment, light, period, phase

## Abstract

The Munich ChronoType Questionnaire (MCTQ) has now been available for more than 15 years and its original publication has been cited 1240 times (Google Scholar, May 2019). Additionally, its online version, which was available until July 2017, produced almost 300,000 entries from all over the world (MCTQ database). The MCTQ has gone through several versions, has been translated into 13 languages, and has been validated against other more objective measures of daily timing in several independent studies. Besides being used as a method to correlate circadian features of human biology with other factors—ranging from health issues to geographical factors—the MCTQ gave rise to the quantification of old wisdoms, like “teenagers are late”, and has produced new concepts, like social jetlag. Some like the MCTQ’s simplicity and some view it critically. Therefore, it is time to present a self-critical view on the MCTQ, to address some misunderstandings, and give some definitions of the MCTQ-derived chronotype and the concept of social jetlag.

## 1. Introduction

Our daily lives are controlled by at least three “clocks”. The clock we know best is the *Social Clock*, representing *Local Time* (“official” social time reference within a given region/time zone). The *Social Clock* allows interaction with others and being in time for school, work, trains, and planes or shop opening times. *Social Time* is related to the *Sun Clock* that has existed ever since the Earth established its stable rotation around its own axis and its sun. The third clock is the circadian clock—our *Biological Clock*—that controls all levels of physiology, from metabolism to behavior, enabling an internal temporal organization in tune with daily environmental cycles.

The rotation of the Earth has not always produced a 24-hour day. When the first circadian clocks developed around three billion years ago (in single-cell ancestors of today’s cyanobacteria), days on Earth were shorter than 17 h long [1], and have been slowing down approximately 2 milliseconds every century—time scales that easily allow evolution to adapt the biological clocks to the changing day lengths. In contrast, our biology’s evolution certainly cannot keep up with the changes we are making to our *Social Clock*.

Before the introduction of time zones in the late 19th century, the *Social Clock* was in synchrony with the *Sun Clock*; noon was close to when the sun stood in its zenith and midnight was 12 h later, halfway between dusk and dawn. With the establishment of time zones, noon became a more artificial concept that only corresponded to the time of the *Sun Clock* on the meridian that defines the respective time zone. The difference between the *Sun Clock* and the *Social Clock* was meant to be no more than 30 min but, dependent on time zone assignments, it can be much more: Galicia in north-western Spain is 1 ½ hours out of synch with the sun and in China, which uses a single time zone despite its huge longitudinal range (73–135 °E), the difference can be more than four hours.

*Biological Clocks* need environmental cyclic signals (zeitgebers) to synchronize to the cyclic environment. The main zeitgeber for the clocks in most organisms is appropriate cycles of light and darkness. A light–dark zeitgeber is “appropriate” for humans, as for other animals, if the duration of its light portion (photoperiod) or its corresponding dark portion (scotoperiod) is not too short, if the light–dark cycle’s period length is close to 24 h, and if the intensity difference between the photo- and the scotoperiod is strong enough (zeitgeber strength). The most common zeitgeber for humans is the natural day’s sunlight and the natural night’s darkness, but theoretically, all other light–dark cycles can serve as zeitgebers as long as they follow the above criteria of appropriateness.

### Principles of Entrainment

When *Biological Clocks* actively synchronize (entrain) to light–dark cycles, they not only show the same period as the zeitgeber cycle (on Earth, 24 h), but also establish a stable relationship with the zeitgeber, called the phase of entrainment. Due to genetic variance, the protein components of *Biological Clocks* can differ between individuals, so that different people may synchronize differently to the same light–dark cycle—earlier or later, the colloquial larks and owls. Inter-individual differences in this phase of entrainment, also called chronotypes, are most likely due to a combination of how the individual clocks respond to light and darkness and how long an internal day they produce [2]. If a *Biological Clock* produces days that are slightly shorter than 24 h, then it has to be entrained differently than a clock that produces internal days that are slightly longer than 24 h; in the former case, the internal day has to be delayed or lengthened, and in the latter, it has to be advanced or shortened.

As described above, the *Social Clock* was historically phase-consistent with the *Sun Clock* (external consistency), as well as consistent with the *Biological Clock* (internal consistency). While in modern, industrialized societies *Local Time* obviously remains socially consistent, it has lost both its external and internal consistency. The external inconsistency was augmented by the introduction of daylight saving time, which simply advances social timing with little influence on biological timing. The internal inconsistency was inflated by a weakening zeitgeber strength [3]: shielding ourselves from daylight by living predominantly in buildings throughout the day and illuminating the night with artificial light have greatly weakened the zeitgeber’s strength and the artificial light in the evening has delayed the *Biological Clocks*, thereby greatly widening the difference between early and late chronotypes within a population [3,4]. At the same time, these conditions have also greatly increased the difference between the *Social Clock* and the individual *Biological Clocks*, known as social jetlag [5].

Since practically all functions in our body are directly or indirectly organized by the circadian clock, the growing temporal inconsistencies become problematic when we need to consider individual internal time in research or medicine (from diagnosis to treatment). We therefore need ways to assess individuals’ phase of entrainment. In order to provide a quick, cost-effective, scalable and non-invasive measure, we developed a simple instrument, the Munich ChronoType Questionnaire (MCTQ), more than 16 years ago [3] to estimate chronotype as phase of entrainment.

## 2. Chronotype

### 2.1. Concept

Chronotype is often conceptualized as a psychological construct or trait [6,7,8]. In this framework, questionnaires assessing diurnal preferences and classifying individuals into types according to a score were developed (e.g., the Morningness-Eveningness Questionnaire, MEQ, [9]). However, considering the growing amount of knowledge on the circadian system and its organization, we believe chronotype should rather be viewed as a biological construct. This view is also in agreement with the term “chronotype” as it was originally proposed in 1974 [10,11]: “an organism’s temporal organization” or “a temporal phenotype”. We like the term “construct” because chronotype actually pertains to the organization of an entire system and not to one of its subparts, like the suprachiasmatic nucleus (SCN) or liver (the “temporal program”, as Colin Pittendrigh called it, [12]). It is thus virtually impossible to directly assess an individual’s phase of entrainment, i.e., her or his internal time, since there is no single circadian phase of entrainment of an organism. The many different oscillators within the organism establish phase relationships with each other and with the external zeitgeber cycle [13,14,15,16]. Estimating the state of a complete system is difficult, but we can use the timing of biological processes under its control as biomarkers for it. In humans, such biomarkers are, for example, acrophase of activity (the peak time of a cosine fit; e.g., [17]) or dim-light melatonin onset (DLMO; e.g., [18]). The Munich ChronoType Questionnaire uses a variable derived from self-reported sleep timing for chronotyping [3,19].

### 2.2. MCTQ-Estimation of Chronotype

The MCTQ core module asks 17 simple questions about sleep and wake behavior, carefully distinguishing between bedtimes and sleep times. These questions address i) bedtime, ii) time spent in bed awake before deciding to turn off the lights (prepare for sleep), iii) how long it takes to fall asleep (sleep latency), iv) wake-up (sleep offset) time, and v) get-up time. The questions are accompanied by iconic drawings that represent each of these stages. Sleep onset is calculated by adding sleep latency to the time of sleep preparation. This set of questions is asked separately for workdays and work-free days. This separation is unique to the MCTQ and turned out to be one of the questionnaire’s most useful characteristics.

The MCTQ uses the midpoint between sleep on- and offset on free days (midsleep on free days, MSF) to assess chronotype. The midpoint of sleep has been found to be one of the most accurate behavioral markers for circadian phase [20]. The choice for using midsleep on work-free days was made in consideration of our modern lifestyles and the clash between the *Biological* and *Social Clock*. We believe that on free days, behavior better reflects an individual’s overall circadian phase since the circadian system is under less pressure to adapt. Think of it as in an analogy to heart rate: heart rate is measured in the resting state when one wants to assess the baseline cardiovascular state. We do not want our measure to be “confounded” by the adaptive response. When assessing a chronotype, we aim for the same: estimating circadian phase when the system is not (or at least less) constrained by social/work obligations.

The *Biological Clocks* of most people are too late to wake up without an alarm clock on workdays. So all but the very early chronotypes accumulate a sleep debt during their workweek, which they compensate for on free days. This sleep debt systematically depends on chronotype—the later MSF, the larger the work-week accumulated sleep debt [19]. Our analyses of the MCTQ database show that subjects predominantly compensate for this sleep debt by sleeping in on free days and not by going to bed earlier. To clean chronotype of the confounder sleep debt, we correct MSF (MSF_sc_ = sleep corrected MSF). For this correction, we first calculate the average sleep duration across the entire week (SD_week_) and then correct MSF by subtracting half of the oversleep. This correction is only applied for people who sleep longer on work-free days than on workdays (SD_f_ > SD_w_):(1)$$If\;SD_{f}\leq \;SD_{w}:\;MSF_{sc}=MSF=\;SO_{f}+\;\frac{\;SD_{f}}{2}$$
(2)$$If\;SD_{f}>\;SD_{w}:\;MSF_{sc}=\;MSF-\;\frac{\left(SD_{f}-\;SD_{week}\right)}{2}=\;SO_{f}+\;\frac{\;SD_{week}}{2}$$
MSF = midsleep on free daysMSF_sc_ = midsleep on free days sleep correctedSD_w_ = sleep duration on workdaysSD_f_ = sleep duration on work-free daysSD_week_ = weekly average sleep duration SO_f_ = sleep onset on work-free days

In shift-workers, work schedules may have an even stronger influence on sleep timing than in “normal” day workers. We therefore developed an adapted version of the MCTQ to estimate their chronotype [21]. It is also based on the timing and duration of sleep on work-free days. Comparing sleep on work-free days after different shifts, sleep on free days following evening shifts was the least affected by the specific shift schedule. The MCTQ^shift^ therefore uses MSF after evening shifts for chronotyping. The MCTQ^shift^ additionally offers conversions for workers whose schedules do not include evening shifts.

Additionally, we recently developed and validated a short version (only six questions) of the MCTQ: the µMCTQ [22]. It has a good correspondence with the MCTQ and its estimation of chronotype correlates with the activity (acrophase) and melatonin (DLMO) phase. This shortened version will be especially useful for large-scale studies that aim to collect extensive data sets from large samples and have to minimize the burden on subjects.

### 2.3. Characteristics of MCTQ-Chronotype

Complex biological qualities vary in a continuous fashion among individuals, taking distribution shapes that are more or less normal within a population. This also holds for MSF_sc_. The distributions of MSF and MSF_sc_ in the MCTQ database (as of July 2017) are shown in Figure 1. As described in the figure legend, color-coding is arbitrary since both MSF and MSF_sc_ are continuous variables (based on local time). Note that the sleep correction of MSF makes the distribution slightly earlier and decreases the over-representation of late chronotypes. While chronotype and sleep need appear to be independent characteristics, the difference in sleep duration between workdays and work-free days is nonetheless chronotype-dependent because of our social schedules. The later the chronotype, the shorter the sleep duration on workdays and the longer it is on free days. Extremely early types, on the other hand, experience a shorter sleep duration on free days and longer sleep duration on workdays [19].

The factors producing the inter-individual differences in chronotype underlying the wide distribution of MSF_sc_ (Figure 1) are likely threefold: genetics (e.g., [23,24,25]), the weak and differing zeitgeber signals (particularly light exposure), and age. The benefit of the large collection of questionnaires in the MCTQ database (≈300,000 entries) is that chronotype can be put into different contexts (e.g., age, sex, urban-rural, different latitudes, cultures, and climates) with unprecedented precision (see below).

Circadian formalisms predict that phase of entrainment will change with zeitgeber strength (amplitude of the light signal, [2]), and indeed, when individuals exchange urban lives (weak zeitgeber signals due to indoor environments and access to electric light) for natural light conditions (strong zeitgeber signals), their sleep timing and DLMO advance significantly and, as predicted, in a chronotype-dependent way [4]. Sleep timing is also earlier in populations with no access to electricity compared to those with access to artificial light [26,27,28,29]. Chronotype, estimated by MSF_sc_ as an indicator of phase of entrainment, also complies with this zeitgeber-strength rule: it is earlier in rural areas than in urban ones [30,31,32].

In modern industrialized societies, people are exposed to more irregular light–dark cycles than in the pre-electrical era. Yet, an influence of the *Sun Clock* on the human *Biological Clock* can still be detected: average chronotype (as assessed by the MCTQ) correlates with the position within a time zone—the further to the East, where *Sun Time* is earlier, the earlier the chronotype [32,33]. The coupling between the *Biological* and *Sun Clock* is tight in rural areas and small towns (exactly replicating the sun’s 4-minute delay per longitude), but is less tight in big cities.

Since chronotype is a product of entrainment, it also depends on day-length (photoperiod) and season. MSF_sc_ is generally earlier under longer photoperiods [34,35], and the timing of sleep on free days during spring seems to track the progression of dawn [36], although Daylight Saving Time adds complexity to this equation. Time of sunrise in winter was also associated with chronotype at high latitudes (59 °N–68 °N), with a decreasing strength from adults to children to adolescents [35].

In addition to genetics and entrainment conditions, chronotype is also highly age-dependent. Cross-sectional analyses of the MCTQ database show that chronotype progressively delays from approximately ten years of age to the end of adolescence (around 20 years old), and then advances until the end of life [37,38]. Interestingly, further analyses of the MCTQ database show an age-dependent relationship between chronotype and light exposure (time spent outdoors as assessed by the MCTQ): this dependency exists in children and adults but is insignificant in adolescents [39]. Age-dependencies in circadian light effects have also been shown for melatonin suppression [40]. Whether entrainment changes are due to developmental differences in physiological light reception or in behavioral light exposure (timing, intensity, and spectral composition) remains to be elucidated. Nonetheless, evidence shows that the phenomenon of adolescents presenting a later circadian phase is observed in other species [41] and also pre-industrial cultures [42].

### 2.4. Discussion

#### 2.4.1. MCTQ-Chronotyping: Pros and Cons

The MCTQ-chronotype and its assessment of sleep phase have been validated against biochemical biomarkers, such as DLMO [22,43,44,45] and cortisol [45], and objective behavioral measures of circadian phase (activity acrophase, and sleep behavior from logs or actimetry) [22,46,47]. They are all significantly correlated with MSF_sc_, as one would expect if they are all valid biomarkers for phase of entrainment (the system state), and thus vary more or less together. The current gold standard marker of phase of entrainment is DLMO [18] measured in blood, urine, or saliva [48]. However, these measurements are expensive and burdensome—involving multiple, well-timed sampling. We also lack toolkits that provide instantaneous results. Although circadian researchers are currently developing methods to assess chronotype with one to two measurements, so far, these still involve methodological hindrances that complicate their use in large-scale studies [49,50,51]. The solution to this problem is therefore currently best achieved by questionnaires.

In contrast to other chronotyping questionnaires, the MCTQ estimates chronotype in local time, allowing for numerous downstream calculations rather than a score developed to classify people into types. It asks for actual behavior and not what time people would choose or prefer to perform their activities had they the opportunity to do so, like, for example, the MEQ does. In a sense, asking for “preferred times” is comparable to using data collected on free days. Therefore, it is not surprising that MCTQ- and MEQ-chronotype show a good correspondence [19,52]. However, the hedonic construct of *preference* in chronotyping can also be problematic, since many people suffering from extreme chronotype in a strictly structured society would actually *prefer* to be more moderate in their temporal behavior.

A limitation of the MCTQ is that all its calculations rely on structured work schedules, which might hinder its use in populations with more flexible schedules or relaxed attitudes towards work times. The same goes for populations whose culture and language do not rely on metric-based concepts of time—like some tribes in Amazonia [53,54]. A second limitation is that sleep timing is not only under circadian control but is also homeostatically regulated [55]. Despite using a simplified view of sleep compensation, the MCTQ chronotype computation therefore corrects sleep timing for sleep debt.

#### 2.4.2. The Stability of a Chronotype: Trait or State?

An important conceptual question that keeps causing headaches and confusion is whether chronotype (as phase of entrainment) represents a personal trait (an attribute of a person—free of situational effects) or a current state (an attribute of a person in a situation and attribute–situation interactions) [56]. An individual’s phase of entrainment under a specific zeitgeber signal could well be thought of as a stable trait. However, since the zeitgeber signal people are exposed to can greatly vary in strength and timing, chronotype in the real world may rather represent a state than a trait—making genetic studies based on real-world data particularly difficult but highlighting the breadth of possible states of the circadian system [57].

This view is not in conflict with chronotype being a biological construct since this system state (phase of entrainment) of the *Biological Clock* should indeed change with entraining conditions. In other words, when chronotype assesses phase of entrainment, it is as stable as its entraining environment. In a recent study, late-types could advance their sleep timing and that of other phase markers by approximately 2 h when following instructions for three weeks that resulted in stronger and more regular zeitgebers [58]. The age-dependent changes in chronotype additionally support the idea of the *Biological Clock* being a dynamic system that continuously adapts to varying internal and external conditions [38].

In summary, we suggest abandoning the notion that chronotype reflects a stable personal trait in favor of it being a state, as is to be expected if it reflects phase of entrainment with its dynamic qualities. After all, even when measures imply a more or less stable psychological trait, like diurnal preference from the MEQ, the scored preference differs between people under different zeitgeber strengths [59,60]. Considering this, estimations of chronotype that reflect a state are closer to what actually happens in real life as well as more useful for understanding the mechanisms that underlie associations between chronotype and health states.

Although we acknowledge the dynamic nature of chronotype, we stress its genetic basis. Although one can advance the phase of entrainment of a late chronotype by appropriate zeitgeber changes [58], he/she will relapse to the late phase as soon as the intervention stops [61].

#### 2.4.3. How is Circadian State Related to Health and Disease?

Late chronotype seems to be associated with an increased likelihood of being a smoker, consuming alcohol, and caffeinated drinks [5], and presenting metabolic alterations [62,63] and clinically significant depressive symptoms [64,65]. However, the mechanisms and the presence of a causal relationship are not clear. We believe that rather than being a late-type, it is the conflict with the time constraints imposed by society that (at least partly) explains those associations. Longitudinal studies and pathway analyses are, nevertheless, still scarce [66]. Further studies investigating how the association between chronotype and health/disease is mediated by circadian misalignment should help in clarifying this matter. “Social jetlag”, another concept put forward by the MCTQ, might facilitate this quest.

## 3. Social Jetlag

### 3.1. Concept

As already described, zeitgebers were drastically weakened with the wide-spread usage of electricity. It allowed us to live in buildings most of the day, excluding us from full daylight, and enabled us to switch on artificial light after sunset. We almost live under constant light conditions, exposing us to darkness only when we sleep (Figure 2). 

This weakening zeitgeber strength has widened the chronotype distribution [3,4] and delayed all chronotypes except for the very early larks, who may even advance under weak zeitgeber conditions. Since the *Social Clock* has not followed the large delays of most *Biological Clocks*, the discrepancy between them has significantly increased. This recent development introduced a new weekly structure, which we first noticed when looking at a large collection of long-term sleep diaries [47]. Many sleep-logs looked like subjects were flying several time zones to the west on Friday evenings and returning on Monday mornings without ever actually travelling (see the example in Figure 3). We therefore called this syndrome *Social Jetlag* (SJL) [5].

When we suffer from travel jetlag, our *Biological Clock* is simply not yet aligned with the light–dark cycle of the destination because its active entrainment mechanism takes about a day for each time zone crossed to adjust. Before this steady-state is reached, the circadian clock as a system [68] and even its parts [16,69,70] are misaligned in reference to the new time zone; the misalignment between different organs and physiological rhythms is most probably the cause of jetlag’s effects on health and well-being. 

We proposed SJL as a concept [5] that describes and quantifies the chronic discrepancy between an individual’s *Biological Clock* and the *Social Clock*. As such, we envisioned SJL as a measure of circadian misalignment. Circadian misalignment is described as an abnormal phase angle difference between two or more rhythms, be they just internal or both internal and external (reviewed by Vetter et al. 2018 [71]). If the *Biological Clock* of a late chronotype is stably entrained to a late phase in the light–dark cycle despite having to get up early with an alarm clock five days a week, one can presume a misalignment between the *Biological Clock* and social schedules. In this case, however, the misalignment is not transient, as it is for travel jetlag, but chronic.

### 3.2. Social Jetlag Computation

Originally, SJL was defined as the absolute difference between the midsleep on free days (MSF) and that on workdays (MSW; see Equation (3)) [5]. However, it is often also informative to use the actual difference: Since negative SJL occurs when midsleep times on workdays are later than those on free days, it may be wise to look at negative and positive SJL separately. Figure 4 shows that the proportion of people who suffer from negative SJL is relatively small. Actual MSF-MSW also makes the distribution of SJL slightly less skewed.
(3)$$SJL=\;\left|MSF-MSW\right|$$
SJL: social jetlagMSF: midsleep on free daysMSW: midsleep on workdays

### 3.3. Characteristics of Social Jetlag

Typically, SJL is greater in late chronotypes (see Figure 5). On workdays, the sleep of late chronotypes is curtailed at both ends of the night: the late circadian sleep window (opening after the wake maintenance zone; [72]) prevents them from falling asleep early enough and the use of an alarm clock prevents them from completing their sleep. On work-free days, the sleep of late chronotypes is ideally free from these external impositions, resulting in later and longer sleep (the oversleep caused by the sleep debt accumulated during the workweek). As a result, MSF is usually later compared to MSW, resulting in SJL. The relationship between SJL and MSF_sc_ is not necessarily linear, since early chronotypes are often forced—by social norms—to stay up later than they would at night, which results in intermediate chronotypes presenting the lowest levels of SJL [5] (see Figure 5). SJL is also positively associated with perceived sleep debt [73], making it difficult to disentangle pure sleep timing effects and those of sleep deprivation.

Similar associations as between age and MCTQ-chronotype are seen between age and SJL [74,75]. Since school start times are often not attuned to the adolescents’ late phases, they experience the most severe SJL, which decreases, but continues to be present, through work life until retirement.

There are several behavioral outcomes associated with SJL: less healthy dietary patterns [76], a higher probability of being a smoker [5], a worse academic performance in high school and university [33,77], and higher physical and verbal aggression in undergraduate students [78]. There is also a great amount of evidence for an association between SJL and risk for metabolic disorders and/or being obese [75,79,80,81,82,83,84]. Depressive symptoms also seem to be associated with SJL [65,85,86], although such a link has not been found with minor or sub-syndromal psychiatric symptoms [87,88], depressive symptoms in young students living in a rural area in Brazil [89], or in healthy controls vs. a clinical sample [90]. Conflicting findings in the literature might be a consequence of methodological heterogeneity, as well as diverse sample characteristics, especially in the case of multifactorial conditions such as neuropsychiatric disorders.

### 3.4. Discussion

#### 3.4.1. What Does Social Jetlag Quantify?

When we conceived the calculation and the term social jetlag, we saw it as a proxy for circadian misalignment—literally quantifying the “discrepancy between social and biological time” [5]. If sleep on free days is indicative of (or close to) a person’s general phase of entrainment (if such even exists), then one can picture sleep times under constraints of working times to be “unnatural” or “against the *Biological Clock*”, and the difference between the unconstrained and constrained sleep times should be a good approximation of “how much one lives against one’s *Biological Clock*”. However, evidence from controlled studies that mirror “real life” and from actual real-life studies has been accumulating that typical physiological circadian phase markers in humans (such as melatonin and cortisol) move in conjunction with advanced or delayed sleep times quite rapidly (e.g., about 45 minutes after only two days) [22,91,92,93,94,95]—be it through changes in external input via light–dark cycle changes or through internal control mechanisms. Since this indicates that the circadian system does not remain in the same state throughout the week but changes with the shifting sleep times, one can assume that the original SJL concept quantifying the discrepancy between the social and biological time may be too simplistic.

While SJL is most probably a good and useful approximation of the amount of strain on the circadian system exerted by social timing constraints, the following question remains: what is it exactly that SJL captures? If we think of sleep as being the only time of darkness that we experience during our industrialized life, then midsleep time is not only the midpoint of sleep but also the midpoint of darkness. Hence, SJL (as the difference in midsleep times between work- and free days) quantifies how much the timing of our light–dark signal (midpoints of darkness) moves between work- and free days.

A moving light–dark cycle brings jetlag back into SJL. Do we hence have to think of SJL as quantifying the amount of jetlag in our weekly routines, i.e., the repeated re-adjustments of our circadian system to new light–dark cycles? The answer is yes and no. This is because, although with SJL we change the timing of our mid-darkness, we do *not* change the actual solar light–dark cycle, which—despite all artificial light—still seems to significantly influence our circadian timing [32,35,96,97]. Hence, SJL possibly quantifies something more akin to shift work—changes in work (or sleep) times with concurrent changes in the light–dark timing before a background of unchanged solar day–night transitions.

#### 3.4.2. Social Jetlag and Sleep Debt

Given the increasing interest in the relationship between SJL and health and disease, it is important to define possible mechanisms behind these associations. Sleep as a marker of circadian phase or system state has to be viewed as confounded by sleep homeostasis, since sleep times are not under the sole control of the circadian system, but are heavily influenced by homeostatic aspects (sleep depth/time awake). In the majority of cases, SJL arises both from differences in sleep timing between work- and work-free days and from the effects of sleep debt accumulated on workdays (oversleep on work-free days results in later MSF).

In an attempt to disentangle the effects of these two factors on SJL, Jankowski proposed an alternative formula to assess SJL that corrects for sleep debt (Figure 6) [98]. He argued that both MSW and MSF are influenced by sleep debt, which means that, e.g., in people with late chronotype (the majority in industrialized societies), MSW is earlier and MSF later than would happen without the homeostatic influences. Therefore, both should be corrected in order to assess the effects of SJL independent of sleep debt. A correction factor for sleep homeostatic influences on MSF has been used right from the start in order to assess chronotype (MSF_sc_) [19]. Jankowski has suggested correcting MSW (MSW_sc_) analogous to MSF_sc_ and using both these corrected measures in the calculation of SJL (Figure 6).

While uncorrected SJL describes the changes in actual sleep timing and thus actual mid-dark as a measure for circadian misalignment, what does SJL_sc_ reflect? Is this measure closer to the circadian strain caused by the changing light–dark signal that is quantified by SJL? Or, does it even reflect the extent to which the circadian system moves under the changing light–dark signal? The jury is certainly still out. Interestingly, after mathematical simplification, SJL_sc_ is the absolute difference between sleep onset on free days and sleep onset on workdays (SO_f_–SO_w_). Do we expect sleep onset to be a good indicator of circadian sleep phase—not influenced by sleep homeostasis but under circadian control, potentially through the wake maintenance zone?

One way to test these two measures and their meaning is by exploiting our extensive MCTQ database, looking at the special case of people that have SJL but show no sleep deprivation and correlate their health and behavioral outcomes to their SJL.

In keeping with the complexity of the phenotype “sleep timing” as a biomarker for phase of entrainment, one should consider conceptually extending the 2-Process-Model of sleep regulation (circadian and homeostatic; [55]) to a 3-Process-Model that includes a social component. The common denominator term *social* should cover any aspects influencing sleep timing to do with societal and work schedules, but also human behavior—from late TV shows and page-turners to peer pressure via social communication channels—that prevent people from sleeping as early as they could/should.

#### 3.4.3. Misunderstandings About Social Jetlag and Conundrums to be Solved

The association between SJL and a higher risk for disorders in multiple systems strongly suggests that irregular sleep timing (or light–dark signal timing) is an important aspect of unhealthy lifestyles. A common misconception, however, is that the health challenge comes from sleeping in on weekends. Recommendations in the lay press go as far as telling people to get up as early on weekends as during their work week. Although weekend recovery sleep is probably not sufficient for preventing all the shortcomings caused by insufficient sleep over the week [99] and may also delay circadian phase [22,91,92,93,94], it nonetheless seems to prevent the worst: a large cohort study recently found that people with a short sleep duration during workdays have a higher mortality rate if they get no catch-up sleep on weekends than if they do [100]. The study unfortunately did not assess associations with sleep timing. In conclusion, it is pivotal to emphasize that SJL and its related outcomes are rather a consequence of constraints imposed by social clocks on workdays than caused by free-day recovery sleep.

Although multiple findings point to a relationship between SJL and health issues, there is still no consensus about the associations reported. As recently suggested by a systematic review [101], conflicting findings might be a consequence of methodological heterogeneity. Considering time-in-bed as time-spent-asleep, for example, is a common confusion: four out of the 26 studies selected for full-text reading in the review (~15%) used bedtime or time spent in bed to compute SJL. Additionally, two other studies used MSF_sc_ instead of MSF when computing SJL.

Different results might also be a consequence of varying sample characteristics, especially in the case of multifactorial conditions. In fact, most studies investigating the associations between SJL and health are cross-sectional. Further longitudinal studies are needed not only to confirm causal associations but also to clarify under how much and for how long one needs to be exposed to SJL for its consequences to show.

## 4. Outlook

From the beginning, the MCTQ has been accessible online, presenting the possibility to collect data from more than 300,000 people all over the world (it has been translated into 13 languages). The MCTQ has been widely used over the past 15 years across many different fields of research. The questionnaire has been successful because it provides a quick, cost-effective, and accurate way of measuring circadian features that have been correlated with several aspects of human health and performance. The formulas developed in the framework of the MCTQ can be potentially extended to other measurements. For instance, chronotype and SJL can also be assessed with actigraphy data (e.g., [84]) and there are also efforts to extend them to other behaviors, as, for example, meal timing (e.g., [102]).

The impact that research on chronotype has had on education and school policies is impressive. Starting in the 1990s, several studies have now shown how high-school students are constantly sleep deprived because of their late chronotype clashing with early school starting times and how this negatively influences their health and performance [103]. Some schools have, as a consequence, delayed their starting times.

Medical research focuses, for example, on optimizing therapies, by treating subjects at a time of day that maximizes the positive and minimizes the negative side effects. This concept is referred to as chronotherapy [104,105]. Despite the solid mechanistic basis, the concept is not broadly exploited in ongoing clinical trials [106]. Correctly estimating chronotype using logistically feasible methods is essential for chronotherapy efficacy.

Assessing an individual’s chronotype can also be implemented to optimize working schedules, for instance, in shift-workers. Studies have shown that sleep improves (is longer) when schedules are organized according to chronotype (e.g., early chronotypes are assigned to early shifts) [107]. Similarly, research on chronotype and time of day can also be extended to any area of human performance (from cognitive to physical) to optimize this.

Analogously to chronotype, assessments of SJL provide a quantitative marker of circadian misalignment that can be used, for instance, during health prevention campaigns to identify people at risk of developing certain diseases.

The dimension of the MCTQ-based chronotype is time-of-day (of MSF_sc_), which—unlike a score-based assessment—can be used as a reference for designing experiments, for performing analyses (based on internal time rather than external time, [108]), or for performing diagnoses or applying treatment. While external time (the *Social Clock*) is the same for everyone, internal time varies substantially between individuals. For different chronotypes, 08:00 local time may correspond to 10:00 for the *Biological Clock* of early types and to 06:00 for that of late types, with all the circadian consequences for cognitive performance, mood, or immune function, to name just a few. One way to translate *internal time* to *clock time* is to use “hours after MSF_sc_“. If, for example, a drug should be given around wakeup, this corresponds to 05:00 for an early type (MSF_sc_ = 1:00) and 10:00 for a late type (MSF_sc_ = 6:00). “Hours after MSF_sc_“ also allows for a direct comparison between different individuals.

The success of circadian clock research was based on clear definitions, protocols, and formalisms concerning the investigation of circadian clocks in the laboratory—mainly under constant conditions. The success of translating chronobiological insights into the real world will also rely on the factors of clear definitions, protocols, and formalisms—in this case, they will predominantly concern entrained clocks. As in the early days of clock research, it will be crucial for these factors to evolve. Chronotyping and assessing circadian misalignment are at the heart of real-life human chronobiology and their refinement will contribute to the potential success of taking it to the next level.

## Figures and Tables

**Figure 1 biology-08-00054-f001:**
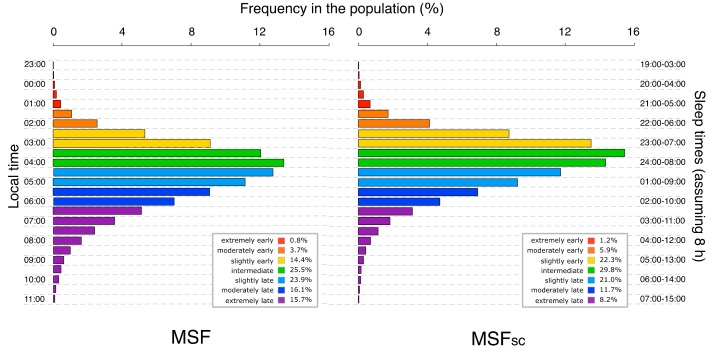
Distributions of midsleep on free days (MSF) (left panel) and midsleep on free days sleep corrected (MSF_sc_) (right panel) in the Munich ChronoType Questionnaire (MCTQ) database (as of July 2017). The distribution is based on half-hourly bins. MCTQ entries were only included in these distributions if all questions of the core-MCTQ were answered, no alarm clocks were used on free days, and values were within a ± 3 σ range. The resulting population sizes were 221,480 for MSF and 185,333 for MSF_sc_ (note that the latter requires information about work status and regular work schedules and is therefore smaller). Color-coding is arbitrary and classifies the population into the seven groups indicated in the legends. The left y-axis shows the local times of the midsleep values, and the right y-axis indicates the sleep window of the respective MSF group (in local time), assuming a sleep duration of eight hours.

**Figure 2 biology-08-00054-f002:**
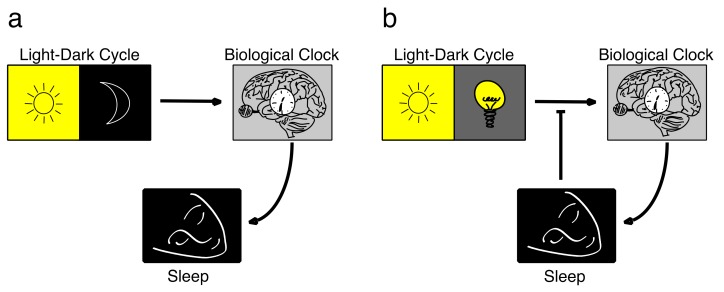
Differences in light conditions for the *Biological Clock* in the pre-industrialized (**a**) and the industrialized (**b**) eras. Historically (**a**), it was the strong difference between natural daylight and darkness that was perceived by the eyes and relayed to the central pacemaker in the suprachiasmatic nucleus (SCN). The SCN neurons entrain to this zeitgeber and transmit this information about day and night to the biological clocks in the rest of the body. Sleep is the major physiological behavior that is under the control of the biological clock interacting with the homeostatic component [67]. The light conditions of the industrialized/urban human environment (see text for details) have resulted in more or less constant light throughout the 24-hour day (**b**), except for the time when we close our eyes during sleep. This situation can also be described as a short circuit between the inputs and the outputs of the circadian system.

**Figure 3 biology-08-00054-f003:**
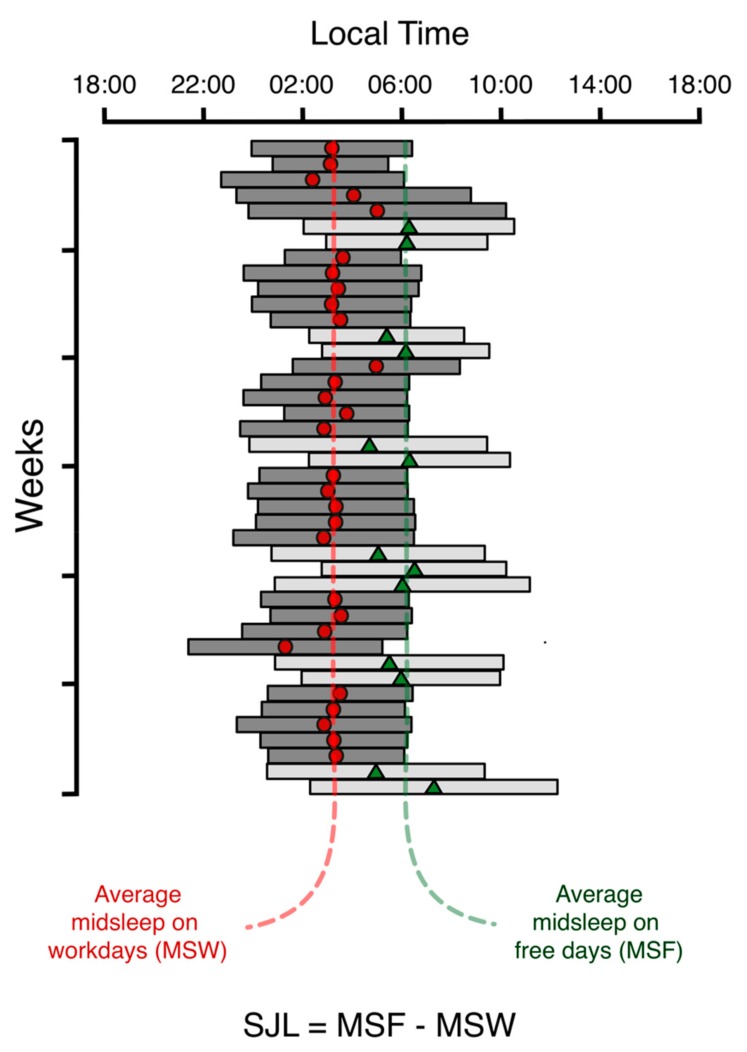
This sleep-log example clearly shows the weekly structure in both sleep timing and duration, which we use as the basis for quantifying social jetlag (SJL). Workday-sleep episodes are dark gray and free day episodes are drawn in light gray. The difference between the average of the midsleep points on workdays (red dots) and those on work-free days (green triangles) is defined as SJL and used as a measure for circadian misalignment (see text for details).

**Figure 4 biology-08-00054-f004:**
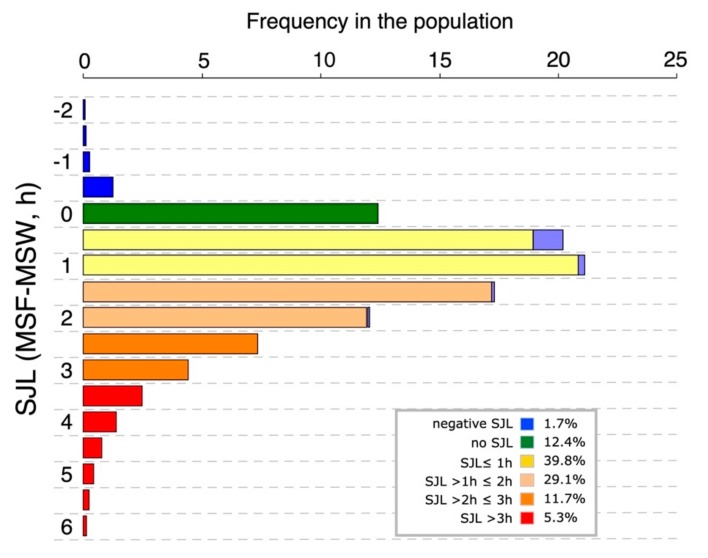
Distributions of social jetlag (SJL) in the Munich ChronoType Questionnaire (MCTQ) database (as of July 2017). The distribution is based on half-hourly bins (population as described in Figure 1 for midsleep on free days sleep corrected (MSF_sc_)). Color-coding is arbitrary and classifies the population into the six SJL groups indicated in the legends. To signify the distribution of the absolute version of SJL (see text for details), the negative SJL categories are mirrored as light blue extensions of the respective positive SJL categories.

**Figure 5 biology-08-00054-f005:**
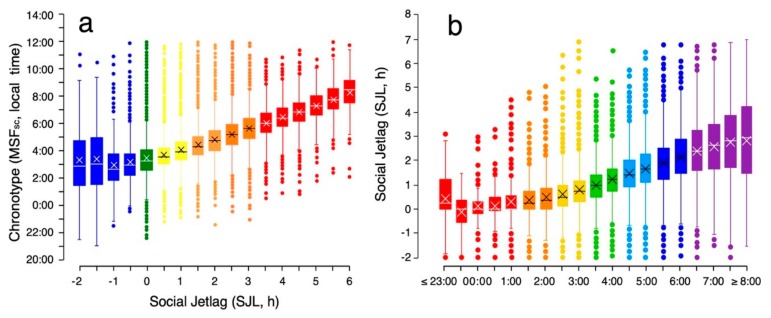
Interrelationship between chronotype (MSF_sc_) and social jetlag (SJL). (**a**) On average, increasing SJL is associated with increasing lateness in chronotype; color-coding is chosen according to the distribution shown in Figure 4. (**b**) Inversely, the later the chronotype, the stronger the SJL. Color-coding is chosen according to the distribution shown in Figure 1.

**Figure 6 biology-08-00054-f006:**
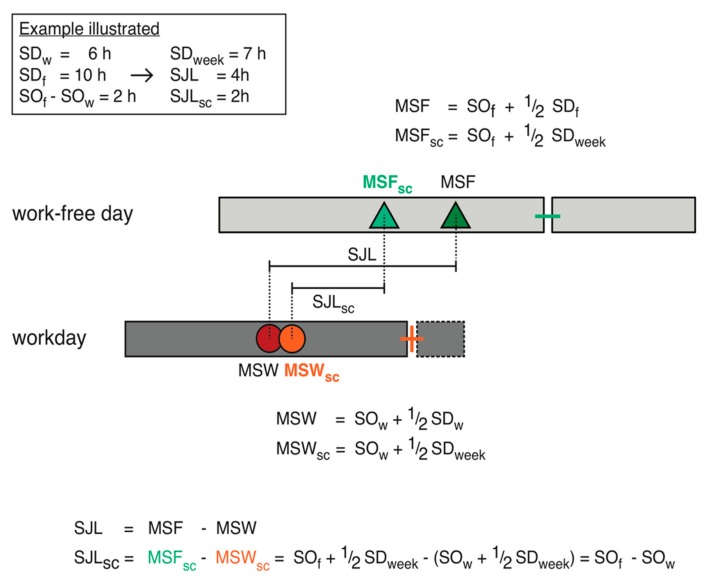
Social jetlag computation. This schematic illustrates the calculations of social jetlag (SJL) [5] and that for social jetlag sleep-corrected (SJL_sc_) as recently suggested by Jankowski [98]. Bars illustrate sleep episodes, as well as their timing and duration, on work- and work-free days, and dots/triangles are the respective midsleep times either including or excluding the dashed parts of the sleep episode. SJL is based on uncorrected, actual midsleep times, thus representing the change of mid-darkness between workdays and free days. SJL_sc_ uses midsleep times that were corrected for a potential oversleep or undersleep in an attempt to remove homeostatic confounders from the sleep schedule. The schematic is drawn to scale and is based on the scenario given in the box assuming a late chronotype with early work schedules in a week with five workdays and two work-free days. Abbreviations: SD_w/f/week_, sleep duration on workdays/on free days/as the daily average across a week; SO_w/f,_ sleep onset on workdays/on free days; MSF, midsleep on free days; MSW, midsleep on workdays; SJL, social jetlag; XX_sc_, sleep corrected.

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
