# Peer review of "Chronotype and Social Jetlag: A (Self-) Critical Review"

_biology, 2019, doi:10.3390/biology8030054_

Round 1

Reviewer 1 Report

The Authors rightly point out that the MCTQ has existed for some years now and that it has a lot of responses recorded, and so feel it is time for a “self-critical” review of the questionnaire. I found the nature of the article itself a bit hard to understand; it seemed to serve as a general introduction to chronotype but then focused almost exclusively on the MCTQ and social jetlag. I wonder if this is better conceptualized as an evaluation of the MCTQ and its application in measuring social jetlag? Certainly, the title needs to include MCTQ.

My main point of issue with this self-critical review is that it is not as self-critical as I was expecting with regard to the MCTQ. The pervasive and widely-held criticism of the MCTQ is that it may not capture the construct of chronotype as well as the MEQ because it evaluates actual timing and not timing preferences. The small paragraph on this issue is probably too dismissive of this concern. This would seem like a good opportunity for the Authors to provide an empirically backed and robust discussion of this issue, to better tease out the subtle differences in this and the state-vs-trait arguments, and when it might be better to use the MEQ or the MCTQ (or when both might be desirable).

Throughout, the language needs to be tempered and tightened to be less subjective (more academic in tone) and to demonstrate more empirical evidence. This would help better state the current understanding and to better differentiate actual findings from potential applications, which on my reading were not infrequently conflated. In many instances, citations and brief explanations would aid the arguments.

Further, specific comments are below.

1.       Given this is almost entirely about one measure, the MCTQ, this needs to be reflected in the title e.g., “a self-critical review of MCTQ usage".

2.       The italicization of key words is so frequent as to be distracting. Unless this is journal style, please remove it, especially as it is also conflated with the Author’s own emphasis such as on lines 87 and 88.

3.       Lines 31 to 36. Citations needed

4.       Line 43. Replace synch with synchronization

5.       Line 45. It’s unclear if the Authors mean to the solar day or to each other

6.       Lines 47 to 50. Doesn’t this also hold true for all other organisms? This would seem to be phrased as a truism. Can the Authors provide some guidance on what the limits of these values are for humans?

7.       Line 55. Remove ‘presently’ as this is not likely to change anytime soon

8.       Line 61. It’s unclear what is meant by “has to be entrained differently...”. Is not the mechanism of entrainment the same, even if the resultant phenotype can be different?

9.       Line 64. Some clarity on what is meant by “consistent” would help. I think the authors are describing that these three clocks were traditionally phase and amplitude locked?

10.   Line 72. Body Clock and Body Clocks seems to be used interchangeably, but the distinction has not been explained. A passage needs to be included early on to orient the reader to the relationship between the central and peripheral oscillators. It seems to be addressed on line 92 but only in passing

11.   Line 91. But wouldn’t you consider the SCN to be the primary oscillator, and so the true (or at least primary) measure of internal time from which to derive a phase angle?

12.   Line 94. Given that the manuscript seems to be written for someone without a lot of expertise in the field, please define acrophase

13.   Line 100. “Literally leading people into and out of bed’ is hard to interpret and reads as if it is a prescription (or intervention) for bed and wake activities. Suggest you rephrase

14.   Line 102. Delete “iconic”

15.   Line 109. “Best” is hard to interpret. Does this refer to e.g. precision, accuracy, ease of use? A little more detail would be helpful

16.   Lines 117 to 126. The rationale for subtracting only half the oversleep is not readily understood. Has the validity of this approach been demonstrated? This should be cited. If people sleep longer during the work week than on free days, would the Authors then consider the MSW to be a better marker than MSF?

17.   Line 140. Please state the measure of reliability between the MCTQ and the uMCTQ

18.   Line 144. Citation needed

19.   Line 146. Although the trends are unlikely to change much, please provide up-to-date data (here and other Figures)

20.   Figure 1. Rotate the figures –90 degrees to aid in its interpretation as this is the way we typically view normally distributed data. Similarly, it would be helpful to overlay the data with a line depicting normal distribution. Why does the Frequency % go from 12 to 1 (should this be 16)? Right y-axis label is clipped. Some explanation of how the 7 groups were generated would be good as these seem arbitrary at this point - what is the empirical/statistical basis?

21.   Line 168. The authors here and elsewhere state 300,000 but show much less in Figure 1?

22.   Lines 171 to 178. Is it possible though that these changes are again due to work pressures (e.g. animal activity, avoiding the heat of the day) rather than directly to a purported stronger light zeitgeber signal? If studies have addressed this, this should be mentioned.

23.   Line 212 onwards. Please quantify the “good” correspondence. More discussion is needed of the MCTQ in relation to the MEQ as this was the original and most widely used scale of chronotype. It’s not particularly clear how the MCTQ is different/better/worse than the MEQ. Is the argument that MEQ is trait based and MCTQ biologically based? If these are truly separate constructs, why then do the questionnaires have good correspondence? The argument on preference being problematic seems like opinion, and in fact the significant and ongoing criticism of the MCTQ is that it does not measure innate preferences, but rather sleep/wake behaviors that are readily modified by societal pressures (and so some would say that the MCTQ is not a measure of chronotype at all). Is there any evidence to suggest that for example that DLMOs are better linked to MCTQ than MEQ chronotype?

24.   Line 215. I don’t believe this assertion is correct. Do the Authors mean that the “best” tool depends on the type of study being performed. No one is likely to argue that a questionnaire is the better (more accurate) solution than DLMO.

25.   Line 227. It’s hard to interpret “Same goes for populations who do not have a clear concept of clock time (as in Social Clock)” and “despite using a simplified view of sleep compensation”. Please provide examples or rephrase

26.   Line 232. “that keeps causing headaches and confusions” is too casual. Delete the phrase or replace with something like “a matter of debate” because this is a critical  conceptual point that should not be so readily brushed aside

27.   Line 231 section 2.3.2. –

28.   The Authors argue for a state rather than trait approach, but why can’t it be a trait that is modulated by environmental factors (as are pretty much all traits)? Chronotype as state does not seem more “realistic” to me than as trait.

29.   If this section is on stability, then an important consideration that is lacking is that there are well described changes in chronotype based on age, including a shift to evening types in adolescence and perhaps morning types in the elderly. This should be incorporated into the state-vs-trait arguments here. What does the evidence support?

30.   Under stable entraining conditions – an example would be good here to contrast to the self-selected conditions. Do you mean in lab vs in the real world?

31.   What is meant by inter and intra-individual phase relationships?

32.   Figure 3. The text on this is very hard to read. Suggest you revise to match the other figures as this seems to be the only data figure that appears to be hand drawn?

33.   Figure 5. Typo on axis labels (“localt”)

34.   Line 362. Quantify or give a range for “quite rapidly”

35.   Line 369. Why should sleep = darkness? For example, people typically don’t rise with the sun, and people sleep rebound on the weekend as shown by the MCTQ. The argument that midsleep on work vs free days is also the midpoint on darkness on these days seems incorrect.

36.   Line 422. Why obesity and substance use?

37.   Line 478. I’m not aware of this having been done with the MCTQ. Please provide citations and more detail. If this is a future, potential, use of the MCTQ, this needs to be stated with more caution (i.e. “could in the future” rather than “can” be used.

38.   Line 485. Typo? (“do you”)

39.   Line 488. The argument here is quite hard to interpret, can it be rephrased or an example given?

Author Response

Reviewer 1
The purpose of peer reviews is to improve the quality of the paper to be published. The
constructive criticisms should be specific to the paper at hand and concrete; sources of
specific critiques and suggestions for improvements should be given.
We have great difficulties with many comments of reviewer 1 since they show little of the
above criteria but instead simply constitute a biased declaration of preference for “diurnal
preference” and a profound dislike for the MCTQ. Many of the comments have no scientific
basis and show a profound misunderstanding of what constructs MEQ and MCTQ try to
assess.
Reviewer 1 writes “The pervasive and widely-held criticism of the MCTQ is that it may not
capture the construct of chronotype as well as the MEQ because it evaluates actual timing
and not timing preferences”. This statement shows the mentioned bias and does not refer
in any way to what the MS actually represents. MEQ and MCTQ are not competing to
capture the same construct as we argue in detail in the MS. So, this cannot even be a
criticism.
The reviewer also writes “…the significant and ongoing criticism of the MCTQ is that it
does not measure innate preferences, but rather sleep/wake behaviors that are readily
modified by societal pressures (and so some would say that the MCTQ is not a measure
of chronotype at all).” This paragraph shows so much confusion about the matter at hand.
Yes, the MCTQ does NOT “measure innate preferences” (we are actually not quite sure if
there is such a thing as “innate preferences”); yes the MCTQ DOES use “sleep/wake
behaviors that are readily modified by societal pressures” to assess phase of entrainment;
that is the heart of the MCTQ!
The recurrent mentioning of “The pervasive and widely-held criticism of the MCTQ” or “the
significant and ongoing criticism of the MCTQ” are general statements without back-up.
Who are these critics? Where is this published? The review process should be scientific
and not clerical. The MEQ is not the original scripture (“MEQ as … the original and most
widely used scale of chronotype”) and the term chronotype was first used in the
conjunction of “phase of entrainment” (referring to a local time) and not with diurnal
preferences (resulting in score). Ehret (1974) defines chronotype as the “temporal
phenotype of an individual/organism”. This definition is in full agreement with our view of
chronotype as a biological construct. See in the figure below that he referred to chronotype
by showing different phase markers.
Ehret, C.F. The sense of time: Evidence for its molecular basis in the eukaryotic geneaction
system. Adv. Biol. Med. Phys. 1974, 15, 47–77.
Horne & Östberg (1976) state in their original MEQ paper: “the aim of the present study
was to further assess the concept of Morningness-Eveningness (…)”. In the paper it is
also stated that the MEQ was re-designed from an original version “to discern individual
differences for suitability to shift work”. In this paper, the term “chronotype” is not
mentioned once.
On the other hand, “chronotype” is in the title of the original MCTQ paper, which has been
the most cited study among research papers including the word “chronotype”, according
to a recent bibliometric analysis (Norbury, 2017). Four of the top 5 most cited papers were
studies using the MCTQ.
Before the MCTQ paper was published in 2003, the keyword chronotype was used in 18
publications, many of which did not refer to the MEQ but to phase of entrainment, e.g., in
rodents. The original MEQ publication used the terms morningness or diurnal preferences.
Before the MCTQ paper was published 145 papers use these two terms. The usage of
chronotype in the context of the MEQ only takes off after the original MCTQ publication in
2003. From the perspective of “only what was originally proposed is correct”, one could
therefore argue that chronotype refers to the biological state phase of entrainment rather
than to the psychological trait diurnal preference. However, since the term chronotype has
also been adopted by the MEQ world, we made the distinction between the biological
construct, which refers to phase of entrainment (regardless whether measured by a
questionnaire, by actimetry, physically [e.g., temperature], or biochemically [melatonin,
cortisol, etc.]), and the psychological construct of diurnal preference, which may
encompass other psychological aspects that influence behaviour. We discuss these
differences at length in the manuscript (see section 2.1).
We have tried to incorporate the unbiased recommendations of this reviewer and we have
provided reasons for not doing so when the criticisms had no scientific background but
were rather based on personal opinions.
The Authors rightly point out that the MCTQ has existed for some years now and that it has a
lot of responses recorded, and so feel it is time for a “self-critical” review of the questionnaire. I
found the nature of the article itself a bit hard to understand; it seemed to serve as a general
introduction to chronotype but then focused almost exclusively on the MCTQ and social jetlag.
I wonder if this is better conceptualized as an evaluation of the MCTQ and its application in
measuring social jetlag? Certainly, the title needs to include MCTQ.
My main point of issue with this self-critical review is that it is not as self-critical as I was
expecting with regard to the MCTQ. The pervasive and widely-held criticism of the MCTQ is
that it may not capture the construct of chronotype as well as the MEQ because it evaluates
actual timing and not timing preferences. The small paragraph on this issue is probably too
dismissive of this concern. This would seem like a good opportunity for the Authors to provide
an empirically backed and robust discussion of this issue, to better tease out the subtle
differences in this and the state-vs-trait arguments, and when it might be better to use the MEQ
or the MCTQ (or when both might be desirable).
Throughout, the language needs to be tempered and tightened to be less subjective (more
academic in tone) and to demonstrate more empirical evidence. This would help better state
the current understanding and to better differentiate actual findings from potential applications,
which on my reading were not infrequently conflated. In many instances, citations and brief
explanations would aid the arguments.
Further, specific comments are below.
1. Given this is almost entirely about one measure, the MCTQ, this needs to be reflected in
the title e.g., “a self-critical review of MCTQ usage".
This review goes well beyond just describing what can be measured with the MCTQ
(chronotype and social jetlag). We have extensively discussed their meaning,
characteristics, usage and relevance from different perspectives. For instance, chronotype
as an estimation of phase of entrainment (PoE) can just as well be assessed by objective
measures (actimetry, biochemical measurements).
2. The italicization of key words is so frequent as to be distracting. Unless this is journal style,
please remove it, especially as it is also conflated with the Author’s own emphasis such as on
lines 87 and 88.
We will keep the formatting as it is.
3. Lines 31 to 36. Citations needed
Thank you for drawing our attention to that. We added a ref and corrected the sentence,
since days were even shorter than 22h 3 billion years ago.
“When the first circadian clocks developed something like three billion years ago (in singlecell
ancestors of today’s cyanobacteria), days on Earth were shorter than 17 hours long
[1], (...)” – line 33
4. Line 43. Replace synch with synchronization
No reason to change, we prefer the generally used term “out of synch”.
5. Line 45. It’s unclear if the Authors mean to the solar day or to each other
We have added “to the cyclic environment” to make the statement clearer.
“Biological Clocks need environmental cyclic signals (zeitgebers) to synchronize to the
cyclic environment.” – line 45
6. Lines 47 to 50. Doesn’t this also hold true for all other organisms? This would seem to be
phrased as a truism. Can the Authors provide some guidance on what the limits of these values
are for humans?
We have added “as for other animals” to make the statement clearer.
“A light-dark zeitgeber is ‘appropriate’ for humans, as for other animals, if the duration of
its light portion (photoperiod) or its corresponding dark portion (scotoperiod) are not too
short, if the light-dark cycle’s period length is close to 24 hours, and if the intensity
difference between the photo- and the scotoperiod are strong enough (zeitgeber
strength).” –line 47
7. Line 55. Remove ‘presently’ as this is not likely to change anytime soon
Done.
8. Line 61. It’s unclear what is meant by “has to be entrained differently...”. Is not the
mechanism of entrainment the same, even if the resultant phenotype can be different?
We have added “in the former case the internal day has to be delayed or lengthened. In
the latter advanced or shortened” to make the statement clearer
9. Line 64. Some clarity on what is meant by “consistent” would help. I think the authors are
describing that these three clocks were traditionally phase and amplitude locked?
We have added phase-consistent. We are not sure what the reviewer refers to when
mentioning the amplitude of social clocks.
“As described above, the Social Clock was historically phase-consistent with the Sun
Clock (external consistency) (...)” - line 66
10. Line 72. Body Clock and Body Clocks seetms to be used interchangeably, but the
distinction has not been explained. A passage needs to be included early on to orient the reader
to the relationship between the central and peripheral oscillators. It seems to be addressed on
line 92 but only in passing.
One is the plural of the other and used when we refer to that system in more than one
individual. Definition is given in line 29. We changed “body clock” for “biological clock” as
suggested by reviewer 3.
11. Line 91. But wouldn’t you consider the SCN to be the primary oscillator, and so the true (or
at least primary) measure of internal time from which to derive a phase angle?
We believe that this definition is in view of the many clocks in the body too rigid and
hierarchical. Translation of chronobiology into medicine will depend on a much more
flexible view of the system.
12. Line 94. Given that the manuscript seems to be written for someone without a lot of
expertise in the field, please define acrophase
The definition of acrophase (the phase of the maximum of a least square fitted cosine
curve) has been added as footnote. Providing the definition in the text would take away
from the flow of this sentence. We leave the decision to the Editor whether to keep the
footnote or not.
13. Line 100. “Literally leading people into and out of bed’ is hard to interpret and reads as if it
is a prescription (or intervention) for bed and wake activities. Suggest you rephrase
We rephrased as follows:
“…carefully distinguishing between bedtimes and sleep times.” – line 103
14. Line 102. Delete “iconic”
Since we use icons, we don’t see the necessity to remove “iconic”.
15. Line 109. “Best” is hard to interpret. Does this refer to e.g. precision, accuracy, ease of
use? A little more detail would be helpful
The term “best” has been changed to “most accurate”. – line 112
16. Lines 117 to 126. The rationale for subtracting only half the oversleep is not readily
understood. Has the validity of this approach been demonstrated? This should be cited. If
people sleep longer during the work week than on free days, would the Authors then consider
the MSW to be a better marker than MSF?
This has been described at length in many publications, which are all cited in this MS. It is
actually not true that we are “subtracting only half the oversleep”. We subtract the entire
oversleep but then the calculation of midsleep needs half of the sleep duration.
The majority of the people sleep longer on free days, not during the workweek. The few
that do sleep longer on workdays than on free days will have a negative SJL. The issue is
discussed in the section “Social jetlag and sleep debt”.
17. Line 140. Please state the measure of reliability between the MCTQ and the uMCTQ
We added the following to the text:
“(…) the μMCTQ. It has good correspondence with the MCTQ and its estimation of
chronotype correlates with activity (acrophase) and melatonin (DLMO) phase.” – line 145
18. Line 144. Citation needed
These distributions are presented here for the first time.
19. Line 146. Although the trends are unlikely to change much, please provide up-to-date data
(here and other Figures).
The collection into this MCTQ database has been stopped in 2017. The data provided are
therefore the most up-to-date data we have.
20. Figure 1. Rotate the figures –90 degrees to aid in its interpretation as this is the way we
typically view normally distributed data. Similarly, it would be helpful to overlay the data with a
line depicting normal distribution. Why does the Frequency % go from 12 to 1 (should this be
16)? Right y-axis label is clipped. Some explanation of how the 7 groups were generated would
be good as these seem arbitrary at this point - what is the empirical/statistical basis?
We have tried many different versions and prefer this rotation because it allows to directly
compare the 2 distributions of MSF and MSFsc and to appreciate how the MSFsc
distribution is slightly earlier and with less extremely late chronotypes compared to the one
of MSF.
The labels were clipped and read well now. Thank you for pointing this out.
Any categorization of a continuous variable is arbitrary. Many studies use different number
of categories. The seven categories shown here were originally created to facilitate
feedback to the general public filling in the MCTQ online.
21. Line 168. The authors here and elsewhere state 300,000 but show much less in Figure 1?
Not all of the close to 300,00 entries in the database can be used for all calculations. The
legend explicitly describes the reasons for different number that go into the calculations.
22. Lines 171 to 178. Is it possible though that these changes are again due to work pressures
(e.g. animal activity, avoiding the heat of the day) rather than directly to a purported stronger
light zeitgeber signal? If studies have addressed this, this should be mentioned.
There is a citation for humans Wright et al.
23. Line 212 onwards. Please quantify the “good” correspondence. More discussion is needed
of the MCTQ in relation to the MEQ as this was the original and most widely used scale of
chronotype. It’s not particularly clear how the MCTQ is different/better/worse than the MEQ. Is
the argument that MEQ is trait based and MCTQ biologically based? If these are truly separate
constructs, why then do the questionnaires have good correspondence? The argument on
preference being problematic seems like opinion, and in fact the significant and ongoing
criticism of the MCTQ is that it does not measure innate preferences, but rather sleep/wake
behaviors that are readily modified by societal pressures (and so some would say that the
MCTQ is not a measure of chronotype at all). Is there any evidence to suggest that for example
that DLMOs are better linked to MCTQ than MEQ chronotype?
We would like to make clear again that the two questionnaires are not in competition. The
aim of this review is not a comparison between MCTQ and MEQ to determine which one
is better. As mentioned above we think that the two questionnaires assess 2 different
constructs (biological and psychological), which does not exclude that the two constructs
can show good correspondence. Possible reasons for this are already addressed in the
manuscript (“In a sense, asking for “preferred times” is comparable to using data collected
on free days. Therefore, it is not surprising that MCTQ- and MEQ- chronotype show good
correspondence”). Both MCTQ- and MEQ-derived chronotypes correlate well with DLMO
and this has been reported in the literature already (see ref 40 as an example).
24. Line 215. I don’t believe this assertion is correct. Do the Authors mean that the “best” tool
depends on the type of study being performed. No one is likely to argue that a questionnaire is
the better (more accurate) solution than DLMO.
We do not make any statement on which is the “best tool”, as implied. The sentence refers
to questionnaires being currently the “best solution” to the logistical burden posed by other
phase assessments (including DLMO): methods that involve multiple sampling, immediate
processing of samples, controlled conditions, for example. These requirements often
cannot be complied with in field studies. Furthermore, the choice of tool(s) should be based
on research questions and study design.
25. Line 227. It’s hard to interpret “Same goes for populations who do not have a clear concept
of clock time (as in Social Clock)” and “despite using a simplified view of sleep compensation”.
Please provide examples or rephrase
The sentences now read as:
“Same goes for populations whose culture and language do not rely on our metric-based
concept of time - like some tribes in Amazonia [53,54].”
26. Line 232. “that keeps causing headaches and confusions” is too casual. Delete the phrase
or replace with something like “a matter of debate” because this is a critical conceptual point
that should not be so readily brushed aside
The importance of recent outreach endeavours is to get our science out to the lay public.
SRBR has just formed a committee on this matter. We therefore decided to keep this
phrasing.
27. Line 231 section 2.3.2. –
Although we are not sure if that was the suggestion, we changed “–” for “:”.
28. The Authors argue for a state rather than trait approach, but why can’t it be a trait that is
modulated by environmental factors (as are pretty much all traits)? Chronotype as state does
not seem more “realistic” to me than as trait.
Here we use the terms “trait” and “state” meaning “an attribute of a person, free of
situational influences” and “an attribute of a person in a situation and the attribute-situation
interaction”. The existence of dispositional differences is not under debate. However, since
phase of entrainment highly depends on situation, in the sense that it can rapidly change
according to the strength and regularity of the zeitgeber signal, we propose to consider
what we measure in real life a state.
We rephrased parts of the section to make this point clearer. – line 237
29. If this section is on stability, then an important consideration that is lacking is that there are
well described changes in chronotype based on age, including a shift to evening types in
adolescence and perhaps morning types in the elderly. This should be incorporated into the
state-vs-trait arguments here. What does the evidence support?
We added:
“The age-dependent changes in chronotype additionally support the idea of the circadian
system as being dynamic and adaptive to continuously varying internal and external
conditions.” – line 251
30. Under stable entraining conditions – an example would be good here to contrast to the
self-selected conditions. Do you mean in lab vs in the real world?
We rephrased the sentence:
“In other words, when chronotype assesses phase of entrainment, it is as stable as its
entraining environment.” – line 247
31. What is meant by inter and intra-individual phase relationships?
We rephrased the paragraph in order to make it clearer.
32. Figure 3. The text on this is very hard to read. Suggest you revise to match the other figures
as this seems to be the only data figure that appears to be hand drawn?
We fixed the labels.
33. Figure 5. Typo on axis labels (“localt”)
Thank you. The typo has been corrected.
34. Line 362. Quantify or give a range for “quite rapidly”
The extent of DLMOs phase shifts depends on the changes in sleep timing. Significant
phase shifts have already reported after only 2 days (e.g. later sleep during weekends).
This has now been specified in the manuscript. – line 378
35. Line 369. Why should sleep = darkness? For example, people typically don’t rise with the
sun, and people sleep rebound on the weekend as shown by the MCTQ. The argument that
midsleep on work vs free days is also the midpoint on darkness on these days seems incorrect.
In industrialized societies, periods of light and darkness are not anymore only determined
by the rotation of the Earth around the Sun, i.e. we can be exposed to (artificial) light also
before sunrise and after sunset. For this reason, we write that sleep is the only time of
darkness we experience (when we sleep our eye lids are closed and, even if there were
light in the room, very little of it would reach our eyes; see Kantermann and Roenneberg,
Chronobiol Int, 2009). Therefore, the argument that midsleep is also the midpoint of
darkness is correct.
36. Line 422. Why obesity and substance use?
We initially chose those variables because they are in the large MCTQ database.
However, we rephrased the statement to make it more general:
“One way to test these two measures and their meaning is by exploiting our extensive
MCTQ database, looking at the special case of people that have SJL but show no sleep
deprivation and correlate health and behavioral outcomes to their SJL.” – line 438
37. Line 478. I’m not aware of this having been done with the MCTQ. Please provide citations
and more detail. If this is a future, potential, use of the MCTQ, this needs to be stated with more
caution (i.e. “could in the future” rather than “can” be used.
Some of the suggested uses of the MCTQ have already been implemented in the
published literature (e.g. influence of internal time – assessed as hours since MSFsc – on
grades in high-school students; van der Vinne et al. JBR 2015). Some others are
suggestions for the future.
38. Line 485. Typo? (“do you”)
Thank you, it is taken care of.
39. Line 488. The argument here is quite hard to interpret, can it be rephrased or an example
given?
We rephrased this sentence:
“One way to translate internal time to clock time is to use “hours after MSFsc“. If, for
example, a drug should be given around wakeup, this corresponds to 5 am for an early
type (MSFsc = 1:00) and 10 am for a late type (MSFsc = 6:00). “Hours after MSFsc“ also
allows direct comparison between different individuals.” – line 504

Reviewer 2 Report

This review is well written.  A letter article titled “Night shifts: chronotype and social jetlag” has been published (BMJ 2018; 361 doi: https://doi.org/10.1136/bmj.k1666). This article includes several references that are relevant to this review and. It would be nice to mention these. 

Minor:

Page 7, line index 305: MSW needs full description the first time it appears. I notice the spelt-out version is in line index 309. 

Figure 3: Handwriting words, especially words with red and blue colours at the bottom of the graph are not clear.  Please write these more clearly. 

Figure 2: It was difficult for me to understand the figure at first sight. I could not figure out that the line drawing was a face. Does the line from the face indicate inhibitory effects on the line from the Sun and the light? I suggest adding some information in the figure. 

Figure 6 legend, line index 401:  “Grey bars…”. On my computer bars are in dark blue, not grey. 

Author Response

Reviewer 2
This review is well written. A letter article titled “Night shifts: chronotype and social jetlag” has
been published (BMJ 2018; 361 doi: https://doi.org/10.1136/bmj.k1666). This article includes
several references that are relevant to this review and. It would be nice to mention these.
Minor:
Page 7, line index 305: MSW needs full description the first time it appears. I notice the speltout
version is in line index 309.
We added the abbreviation after the first description.
Figure 3: Handwriting words, especially words with red and blue colours at the bottom of the
graph are not clear. Please write these more clearly.
Done.
Figure 2: It was difficult for me to understand the figure at first sight. I could not figure out that
the line drawing was a face. Does the line from the face indicate inhibitory effects on the line
from the Sun and the light? I suggest adding some information in the figure.
Information was added to the figure as suggested.
Figure 6 legend, line index 401: “Grey bars…”. On my computer bars are in dark blue, not
grey.
We removed “grey” to avoid confusions.

Reviewer 3 Report

In this review article, the authors described the Munich ChronoType 80 Questionnaire (MCTQ), which was developed for providing a simple and scalable way to measure the individuals’ “phase of entrainment”. They also elucidated “chronotype” and “social jetlag” derived from MCTQ. Sleep disorders have given a strong impact on our modern society (ex, 30% of the population in the USA complains of sleep disorder), and this review could give an insight into human circadian rhythm research.

There are several points the authors could improve for readers.

Terminology: three clocks; social, sun, and body clocks. Because these words are used for different meanings, the authors should define them more carefully and clearly. What is “social clock”? It seems “biological clock” or “circadian clock” is better than “body clock”.

P2 L46: A light-dark zeitgeber is ‘appropriate’ for humans if the duration of its light portion (photoperiod) or its corresponding dark portion (scotoperiod) are not too short, if the light-dark cycle’s period length is close to 24 hours, and if the intensity difference between the photo- and the scotoperiod are strong enough (zeitgeber strength).

> What is the base of this explanation? Please add references.

P2 L56: phase of entrainment > Please show the references.

P2 L66: Local Time > the definition of Local Time is not clear so that the argument in this section is hard to understand.

P4 Fig.1: X-axis of the graph; 0, 4, 8, 12, and 14. Y-axis; Sleep times (assuming 8 hours)

P5 L208: more of less > more or less

P6 L231: 2.3.2. The stability of chronotype – state or trait? > The terms “state” and “trait” are not clear, what is the difference? The authors could add clear explanations or citations.

P7 L295: the circadian clock [interacting > the circadian clock interacting

P9 Fig. 5 (a): What is the unit of y-axis? If it shows the local time, it should be 1:00 am or something I think.

P11 L429: 3.4.3 > what does this mean?

p13 L485: do you > what does this mean?

Author Response

Reviewer 3
In this review article, the authors described the Munich ChronoType 80 Questionnaire (MCTQ),
which was developed for providing a simple and scalable way to measure the individuals’
“phase of entrainment”. They also elucidated “chronotype” and “social jetlag” derived from
MCTQ. Sleep disorders have given a strong impact on our modern society (ex, 30% of the
population in the USA complains of sleep disorder), and this review could give an insight into
human circadian rhythm research.
There are several points the authors could improve for readers.
Terminology: three clocks; social, sun, and body clocks. Because these words are used for
different meanings, the authors should define them more carefully and clearly. What is “social
clock”? It seems “biological clock” or “circadian clock” is better than “body clock”.
We changed “body clock” for “biological clock” throughout the manuscript.
P2 L46: A light-dark zeitgeber is ‘appropriate’ for humans if the duration of its light portion
(photoperiod) or its corresponding dark portion (scotoperiod) are not too short, if the light-dark
cycle’s period length is close to 24 hours, and if the intensity difference between the photo- and
the scotoperiod are strong enough (zeitgeber strength).
> What is the base of this explanation? Please add references.
We added: Williams, G.E. Geological constraints on the Precambrian history of Earth’s
rotation and the Moon’s orbit. Rev. Geophys. 2000, 38, 37–59.
P2 L56: phase of entrainment > Please show the references.
We added: Roenneberg, T.; Daan, S.; Merrow, M. The art of entrainment. J. Biol. Rhythms
2003, 18, 183–194.
P2 L66: Local Time > the definition of Local Time is not clear so that the argument in this section
is hard to understand.
Local time refers to the “official” social time reference within a given region/time zone. This
was also added to the manuscript – line 25
P4 Fig.1: X-axis of the graph; 0, 4, 8, 12, and 14. Y-axis; Sleep times (assuming 8 hours)
Thanks for pointing out that the labels were clipped in this graph. The labels read well now.
P5 L208: more of less > more or less
Thank you. Corrected.
P6 L231: 2.3.2. The stability of chronotype – state or trait? > The terms “state” and “trait” are
not clear, what is the difference? The authors could add clear explanations or citations.
We added the definitions:
“An important conceptual question that keeps causing headaches and confusions is
whether chronotype (as phase of entrainment) represents a personal trait (an attribute of
a person – free of situational effects) or rather a current state (an attribute of a person in
a situation and attribute-situation interactions).” – line 238
P7 L295: the circadian clock [interacting > the circadian clock interacting
Thank you. Corrected.
P9 Fig. 5 (a): What is the unit of y-axis? If it shows the local time, it should be 1:00 am or
something I think.
The unit is local time expressed on a 24-hour scale. The same time unit has been used
also in Figure 1. It is now shows in the figure as hh:mm.
P11 L429: 3.4.3 > what does this mean?
We are not sure what the question is about. If the reviewer refers to the word “conundrum”,
here is the definition: a confusing and difficult problem or question.
p13 L485: do you > what does this mean?
Thank you. Typo corrected.
